# Combining the Advantages of Different Processing Solutions Using a Novel Motion Processing Approach

**Olaf Holowenko** [1,*] , **Clemens Troll** [2] , **Steffen Ihlenfeldt** [1,3] **and Jens-Peter Majschak** [2]

1. Institute of Mechatronic Engineering, Technische Universität Dresden, 01069 Dresden, Germany; steffen.ihlenfeldt@tu-dresden.de
2. Institute of Natural Materials Technology, Technische Universität Dresden, Bergstraße 120, 01069 Dresden, Germany; clemens.troll@tu-dresden.de (C.T.); jens-peter.majschak@tu-dresden.de (J.-P.M.)
3. Fraunhofer IWU, 01187 Dresden, Germany
* Correspondence: Olaf.Holowenko@TU-Dresden.de

**Abstract:** In processing machines, technological tasks are implemented using suitable processing solutions. Those processing solutions can in turn have very different characteristics and specific advantages and disadvantages, e.g., concerning sensitivity to changing operating speed. In state-of-the-art processing machine controls, executing one single processing solution is supported. The execution of various processing solutions together and the combination of their advantages is currently not supported at all. In this article, a motion control approach is discussed that allows combining seemingly incompatible process solutions for a given technological task into a hybrid process solution, using the example of processing machines. The objective of this approach is to increase the achievable process window of the machine in terms of operating speed. It is shown that combining different process solutions can merge their advantages and compensate for their disadvantages. The article brings together the lessons learned from previous work in a new application to exploit advantages and compensate for disadvantages.

**Keywords:** operating-speed dependency; motion control; processing machines; dynamic process control; hybrid process solution



## 1. Introduction

Processing machines are used for the automated production of mass consumer goods in high quantities. Cyclic, dynamic processes at high operating speeds are characteristic for this type of machine. The operating speed indicates how many processing cycles are executed in a unit of time [1]. From the economic point of view, the highest possible operating speed with low downtimes is required.

Processing machines usually do not work independently but consist of several concatenated machines and buffers in processing lines [2]. If a standstill occurs at a single machine, e.g., due to problems in the process, the surrounding buffers continue to be filled or emptied by the machines adjacent to them. After the problem has been solved, the machine continues working. During troubleshooting, it must be ensured that the surrounding buffers neither fill up completely nor are completely emptied. To achieve this, on the one hand, it can be useful to reduce the operating speed of the surrounding processing machines. On the other hand, the operating speed of the faulty machine could be increased after the problem is solved to level out all surrounding buffers. As a result, the operating speed of the individual machines varies. This, in turn, results in the requirement to be able to operate the process from standstill to the highest operating speeds [3]. Furthermore, there is the requirement to be able to change the operating speed while executing the process.

Intra-machinery transportation processes are widely used in processing machines. In a multi-function machine such as a packaging machine, they connect different production

operations with each other. A transportation process often found in processing machines is conveying small-sized goods such as chocolate bars from one rest position to another, e.g., the process described in [4]. One strategy for realizing this type of transportation process is conveying the products along a horizontal path using a rise-to-dwell motion, e.g., the process described in [5]. This motion can be implemented using different processing solutions. In the context of this work, a processing solution is an implementation of the process using specific physical principles. Two possible solutions will be discussed later.

To successfully implement a process solution for an operating speed, given requirements must be met [6], e.g., concerning the achievable motion accuracy of the working tool or product, the product quality, or the environmental impacts, such as vibrations or noise. It is well known from industrial practice that these requirements often cannot be met if the single process solution is executed at varying operating speeds [7]. This becomes particularly apparent when trying to increase the operating speed for economic reasons. Undesirable effects could include non-compliance with the required motion accuracy, reduction in the product quality, or increase in environmental impact. As a result, the applicability of an process solution depends on the given operating speed.

Different process solutions show different reactions to varying operating speed. Because of that, it seems to be reasonable to combine complementary process solutions in order to ensure processing capability over the required large operating speed range. However, such an approach must be supported by the control system. Using conventional control systems in processing machines, the implementation of only one single process solution is supported; the combination of multiple process solutions is not provided by current control approaches.

In this work, an approach is presented with the help of which different processing solutions can be combined and executed together. As shown in the following, process windows of different process solutions for one technological task may overlap under certain circumstances. In this case, a combination of two process solutions would be a great advantage. The objective of this article is the enlargement of the process window in which all given requirements are fulfilled sufficiently. This enlargement can be used to increase the maximum achievable operating speed.

The work is structured as follows: Section 2 summarizes the characteristics of the process and introduces the experimental setup. In Section 3, a conventional solution for this process is discussed. In the following Section 4, an operating-speed-dependent motion processing approach is presented that can be used to achieve higher output rates than possible with the conventional approach. However, the novel transportation solution used here has the disadvantage that it is limited to lower operating speeds. It is therefore deduced at the end of Section 4 that it is reasonable to combine this new approach with the conventional one. Section 5 presents an idea of how to use this motion processing approach for combining the advantages of both process solutions and reducing the disadvantages of the individual ones.

## 2. Example Process under Consideration

### 2.1. Characteristics of the Process

Using a rise-to-dwell motion is one famous strategy for solving a transportation task in processing machines. Two requirements for a rise-to-dwell motion for conveying small-sized goods are given. On the one hand, the rest positions of the product must be known. On the other hand, the product needs to be at standstill in those rest positions. Characteristic for this rise-to-dwell motion is that only one part of the motion profile is relevant for the transportation of the product, the process stroke. The remaining motion, the return stroke, is not relevant as long as it does not influence the process in a negative way.

In this process, the products need to be conveyed between the two rest positions as quickly as possible and without damage. The performance of the process is evaluated based on the operation speed $f$ in Hz and the number of undamaged products handled within a time interval [8]. Additional evaluation criteria can be derived from process

requirements, such as undercutting an allowed positioning error of the product on the rest position. The permitted positioning error depends on the process under consideration and has to be determined individually for each process.

### 2.2. Experimental Setup

Generally, classical linkages are combined with modern servo drives and control systems to implement such a rise-to-dwell motion. Thereby, often one-dimensional mechanisms are used to realize those motions, e.g., in [9]. On the one hand, these mechanisms have various advantages, e.g., in terms of motion accuracy, mechanical limitation of the working space, or synchronous motion [10]. Actuating them by modern servo drives offers further advantages, especially concerning the flexibility to change the motion profile quickly [11,12]. On the other hand, using one-dimensional mechanisms has the disadvantage that only one process solution can be implemented, because the motion specification is given by the mechanism.

In this work, different processing solutions are to be implemented with the same test rig. A two-dimensional setup is used for this purpose, shown in Figure 1. The assembly is a five-bar linkage with two degrees of freedom, driven by two servo motors. The working tool is shaped as a comb with several tines that conveys several products in one process stroke. Since the products handled in the machine are natural products with occasionally large tolerances, the spacing between the tines is to be greater than the width of the product. This setup will be used in the following examples.

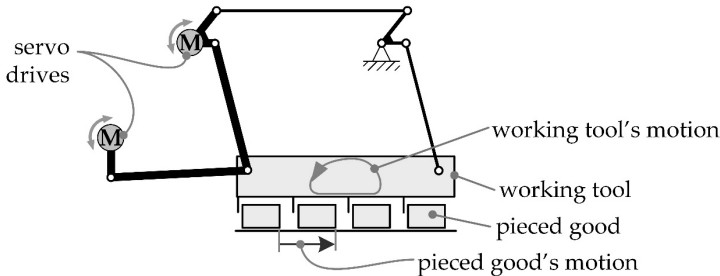

**Figure 1.** Experimental setup for the intermittent transport of pieced goods along a horizontal path [13].

The motion profile for a given process solution is the input for the motion processing. Each motion profile has the character of a one-dimensional path of the comb. This path must be processed with a given operating speed, which is given by a master control system. The set-points for the two servo drives are derived from the geometric parameters of the assembly and the comb's motion path. In order to follow as precisely as required, the motion set-points on the drives have to be processed synchronously. Since this work is about the combinability of different solutions, kinematic aspects and synchronization will not be discussed further.

### 3. Conventional Processing Solution

The conventional transportation solution, which is widely used in industry, is illustrated in Figure 2. In this example, the product is shoved by the comb's first tine over a sliding surface. Figure 2a shows the process-relevant motion sequence in detail. At the beginning (I), the product is in rest at a starting point. In phase (II), the product does not lose contact with the tine. It slows down by friction with the sliding surface when the working tool decelerates. Phase (III) shows the final position of product and comb. Afterwards, the comb is then lifted and moves back to the starting point of the motion. Since the product always stays in touch with the first tine, the product's transport distance is identical to the comb distance. From the speed diagram in Figure 2b, it can be seen that the comb's speed is also exactly the same as that of the product.

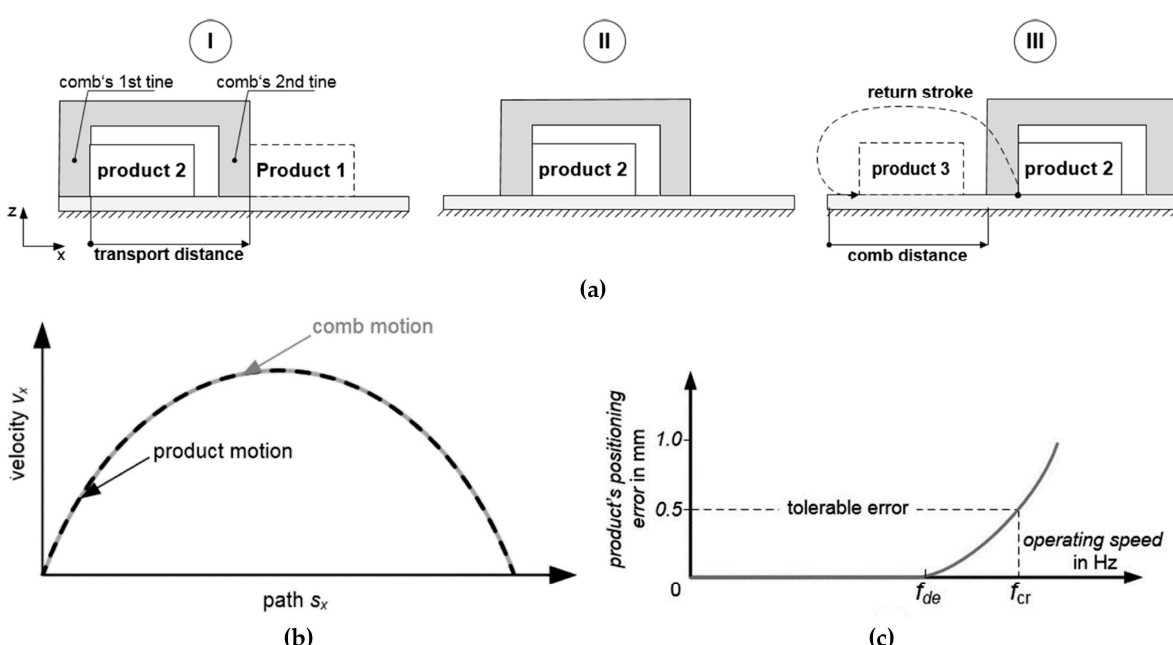

**Figure 2.** Behavior of the conventional process solution. (**a**) Motion principle, (**b**) velocity profile of the working tool and product, and (**c**) trends of the product's positioning error [8].

When the operating speed is increased, e.g., to raise the output of the machine, the kinetic energy of the product increases proportional to the square of the operating speed. Since this energy has to be removed completely by friction, the distance needed to stop the product also increases quadratically. This effect leads to the situation that above a certain operating speed $f_{de}$ (Figure 2c), the product cannot decelerate fast enough to stay in contact with the tine. If the operating speed is increased further, the product becomes more and more insufficiently positioned. At a limiting speed $f_{cr}$, it can no longer meet the positioning accuracy requirements. Both operating speeds, $f_{de}$ and $f_{cr}$, can be raised by various means. For example, the friction used to stop the product can be increased. In industry, mechanical braking elements are often placed above the products. However, this leads to additional stress on the product and can have a negative impact on the product's quality. Regardless of the use of such optimizations, this process solution is limited in terms of speed due to the physical principle used. Investigations in [7] determined experimentally $f_{cr}$ at 1.3 Hz on the available test rig.

The conventional transport process, often found in industry, is limited in its maximum achievable operating speed by its very nature. The reason for this is that, on the one hand, the distance required for stopping the product increases with the operating speed. On the other hand, the available braking distance is limited by the length of the sliding surface. As a result, the operating speed can only be increased up to a certain value. Nevertheless, there is an economic incentive to increase the operating speed. This enables the process to be run at operating speeds from standstill to the highest possible.

## 4. Operating-Speed-Dependent Motion Processing Approach

This chapter summarizes an approach for increasing the operating speed for the rise-to-dwell transportation process that has already been published in preliminary works. The approach consists of three parts, for each of which an overview of the previous research results is given in the following.

### 4.1. Model-Based Optimized Process Solution

In the state of the art, a large amount of literature is available concerning motion profile optimization for automated machines. In this work, only the results of the optimization

(the optimized process solution) are of interest. For the interested reader, we refer to, e.g., the review article [14] or the Ph.D. thesis [15] and the references therein.

The new transport approach discussed in [8] applies a new strategy for implementing the same technical task as the conventional approach mentioned in Section 1. Figure 3a shows the motion principle of tool and product for this process solution in the process stroke. The product is initially (I) located at the same starting point as in the conventional process. At this position, the product is accelerated by the comb's left tine. During phase (II), the comb decelerates (Figure 3b, gray), and the product is entering into a free-sliding phase. While sliding, the product slows down by friction. In phase (III), the comb speeds up again, actively catches the product with the right tine, and stops it at the end point of the motion. The principle of this process solution is based on the concept of partly removing the product's kinetic energy by its friction with the slide surface. The remaining energy is actively absorbed by the second tine.

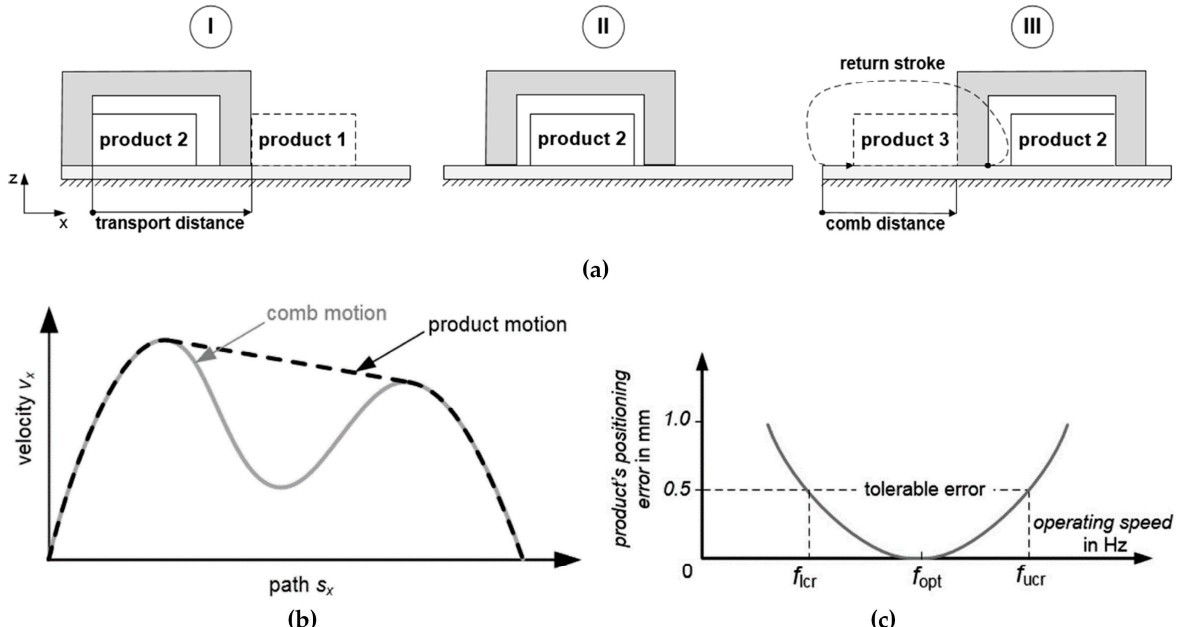

**Figure 3.** Behavior of the new process solution. (**a**) Motion principle, (**b**) velocity profile of the working tool and product, and (**c**) trends of the product's positioning error [8].

In this process, the product is first accelerated and reaches a velocity at which it detaches from the tine. This velocity is proportional to the operating speed. Afterwards, the friction with the sliding surface decelerates the product's motion. The breaking force is independent of the operating speed. In turn, the time available for the product to decelerate decreases with rising operating speeds. Therefore, at higher operating speeds, less product velocity is removed by friction than at lower operating speeds. As a result, the influence of friction on the process decreases with increasing operating speed. Thus, the process is sensitive to changing operating speeds, and the motion has to be optimized operating-speed-dependent for a given velocity $f_{opt}$.

The lowest optimizable operating speed was determined to be at 1.3 Hz in the optimization configuration used in the example. At this speed, the product obtains exactly the kinetic energy from the working tool that is removed by friction on the sliding surface. In this case, actively stopping the product with the second tine is not required. The product's motion is identical to the motion of the conventional solution. At lower operating speeds, the product does not slide far enough to be positioned correctly. On the contrary, the upper limit for the optimal operating speed $f_{opt}$ is mainly determined by the physical limitations of the test rig, e.g., by the achievable motor torques. In [7], operating speeds of 5 Hz were

implemented reliably. This represents a significant increase in operating speed of 285 % compared to the conventional process.

Figure 3c illustrates the operating-speed-dependent positioning error associated with the new process. If the optimized motion specification is executed slower than $f_{opt}$, the product is not accelerated enough to reach the target. From this, a lower critical speed $f_{lcr}$ results. If the product is accelerated too much, it has too much energy and collides with the second tine. In this case, it bounces off and is not positioned as required above an upper critical speed $f_{ucr}$. For the optimal speed $f_{opt}$, the position error is small. When the operating speed falls below $f_{lcr}$ or if it exceeds $f_{ucr}$, the positioning quality requirements are not met. This process is thus limited both upward and downward concerning the operating speed. It was shown in [7] that this process solution is generally suitable for higher operating speeds than the conventional process.

### 4.2. Characteristic Map of Operating-Speed-Dependent Motion Profiles

The second part of the motion processing approach is using several of those operating-speed-dependent optimized motion profiles together to describe the whole process solution. These profiles are combined into a characteristic map of operating-speed-dependent motion profiles. The principle is outlined in Figure 4. This characteristic map has three dimensions: First, the given operating speed $f$ in Hz. The second dimension is a normalized cycle time $\tau$, describing the progress of the motion. Finally, the third dimension is the associated motion set-point.

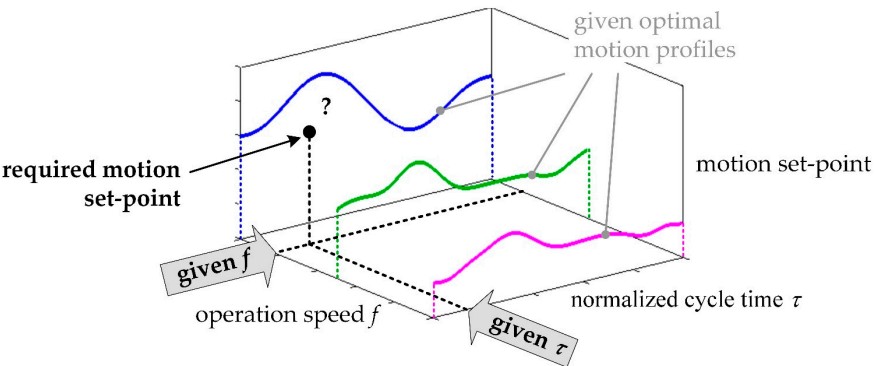

**Figure 4.** Principle of using a characteristic map with three optimized motion profiles [16].

Using this map, a valid motion set-point is to be calculated depending on the two parameters $f$ and $\tau$ (Figure 4, 'required motion set-point'). Thereby, a given operating speed does not necessarily meet any optimized motion profile. Nevertheless, this motion set-point should be as close as possible to the optimal motion profile as it could be optimized. Furthermore, as few optimal motion profiles as possible should be specified to reduce the amount of data. Suitable motion set-points can thus be calculated for many operating speeds.

Using the characteristic map of motion profiles, it should be possible to maintain process stability in a larger process window than with only one motion profile, even with varying operating speeds. Enlarging the process window can increase the machine's maximum output.

### 4.3. Motion Processing Approach

For executing the characteristic map of motion profiles, different approaches were considered and investigated in [16]. The requirement in switching between the optimized motion profiles is that no additional disturbances can be induced into the process. The motion processing approaches can be categorized into discrete and continuous motion processing.

In discrete motion processing, two motion profiles are replaced by each other when the operating speed passes a given switching frequency $f_s$. 'Discrete' thus refers to the handling

of changing operating speeds. Different approaches can be considered for switching [17]. One is switching on a point $\tau_s$ within the motion profiles, as shown in Figure 5a. The prerequisite for using this approach is that a suitable point with special properties exists on two neighboring optimized motion profiles. In this case, all motion parameters (e.g., *s*, *v*, and *a*) have to be equal to achieve $C^2$ continuity when switching between the motion profiles. When using a weighting function (Figure 5b), it is possible to switch between any given motion profiles. A special point for changing the motion profiles is not required. An important requirement for the application of the weighting function is that it needs to be applied in a part of the motion profiles where it is not disturbing the process. In our example, this could be the return stroke. In several works [7,16–19] the discrete processing has been applied and validated. It was shown that the available process window can be increased considerably in certain applications.

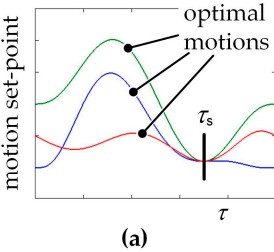 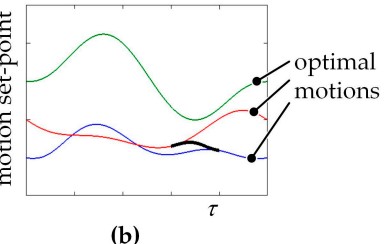

(a)  (b)

**Figure 5.** Principle of the discrete processing of motion profiles. (**a**) Switching on a point. (**b**) Switching using a weighting function. Adapted from [17].

Another approach to switching between optimized motion profiles is continuous motion processing. 'Continuous' means here that with the operating speed continually changing, motion values are interpolated between two optimized motion profiles. In [16,17] it was assumed that this can improve the motion quality compared to the discrete motion processing approach. In [7,13,16] it was experimentally proven that the interpolated value of the continuous processing might be closer to the optimal value compared to discrete processing. However, the success of this depends on the process characteristics and the used interpolation method [13]. In [7], a 1D linear interpolation was validated for the intermittent transport method, shown in Figure 6a. In [19], a novel 2D linear interpolation approach was developed and validated on another process with fundamentally different characteristics, shown in Figure 6b. Using $C^2 \times C^2$ spline surfaces, shown in Figure 6c, was investigated in [16]. This approach was not pursued further due to the higher online calculation effort [16]. Continuous motion processing has been used several times and validated [7,13,16]. Here, it was shown that compared to discrete switching, it can improve motion accuracy particularly [16]. This effect can be used to further extend the process window.

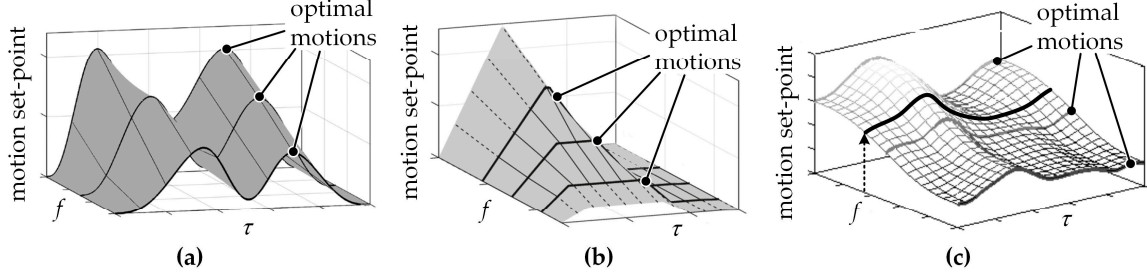

(a)  (b)  (c)

**Figure 6.** Principle of the continuous processing of motion profiles. (**a**) 1D linear interpolation [7], (**b**) 2D adaptive interpolation [19], and (**c**) $C^2 \times C^2$ spline surface [17].

*4.4. Conclusions*

An approach for processing optimized operating-speed-dependent motion profiles has been briefly discussed. Using this motion processing approach, the process window for the intermittent transport of pieced goods can be significantly enlarged, as shown in [7]. The maximum operating speed is also increased significantly. However, the demand to be able to operate from standstill to the highest operating speed and the ability to change the operating speed at the same time are not possible even with the new process solution using a free-sliding product. The reason is its downward limitation in operating speed. This solution thus has considerable advantages, but it also has a disadvantage compared to the conventional process solution discussed in Section 3.

In this context, the idea has arisen to combine the conventional process solution and the newly developed one. The advantages of both solutions should be merged and the disadvantages compensated for. To implement this, however, a motion processing approach is required, with the help of which processing solutions with very different characteristics can be combined.

## 5. Combining Processes with Differing Solution Principles

In this chapter, a new approach for processing different process solutions together is presented. A novel hybrid process solution is discussed using the example of a rise-to-dwell product transportation process. It intends to combine the process capability of the conventional process solution with low-operating-speed support and that of the new approach for higher operating speeds. The advantages of the two solutions are merged, and the process window of the process is increased.

*5.1. Basic Idea*

The basic idea of combinability is based on the knowledge that the technological task ('transport product from position A to position B') is identical for both process solutions. This idea will be discussed as it applies to the trends of the product's positioning error, shown in Figures 2c and 3c. The main principle of combining process solutions is illustrated in Figure 7. The two process solutions used in the figure are colored in gray (conventional solution) and blue (new solution). By overlapping the resulting errors of the conventional process (Figure 7a, limited to higher operating speeds) with those of the new process (Figure 7b, limited to lower speeds), the process window of the resulting process can be increased compared to the conventional process. At the same time, the possibility of operating down to standstill would not be lost. Due to the combination, the conventional process achieves the required tolerance in the lower operating speed range (Figure 7a), while the new process solution covers the upper speed range (Figure 7b). To achieve this effect, the two profiles are placed overlapping on an operating speed $f_s$. The motion processing is performed discretely, and the switching is done at a switching speed $f_s$.

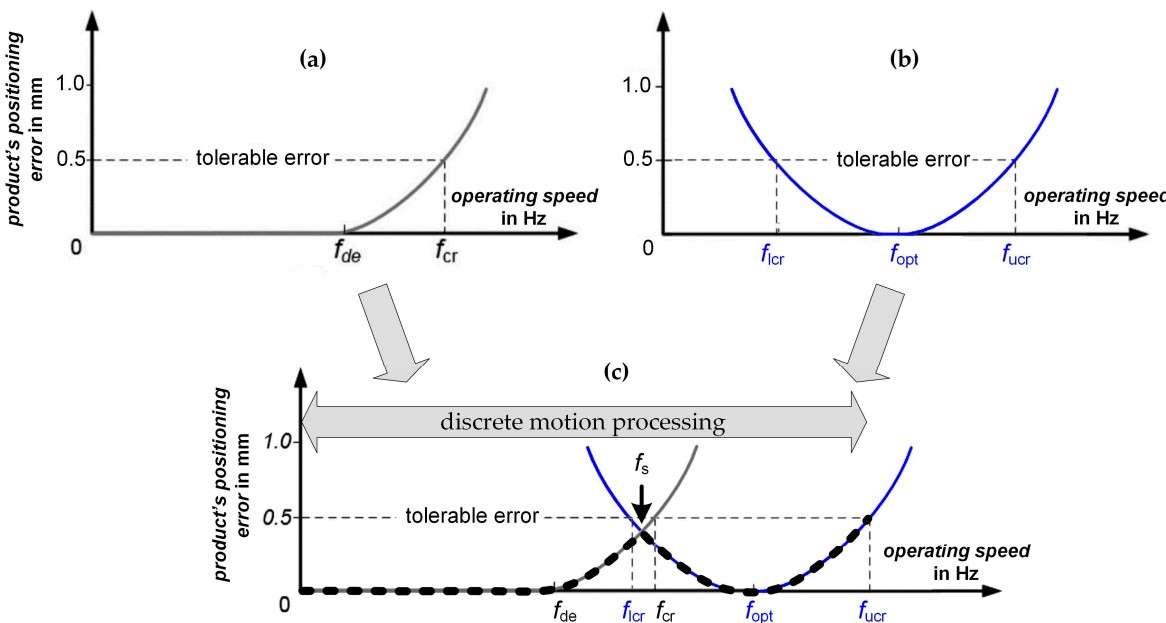

**Figure 7.** Basic idea of the new hybrid processing solution. (**a**) Product's positioning error for the conventional and (**b**) new processing solution. (**c**) Principle of combining the two processing solutions with the aim of increasing the maximum operating speed.

### 5.2. Switching between Motion Profiles for Different Process Solutions

Two process solutions can be combined if they implement the same technological task in relation to the product. In the example processes, the task 'convey product from rest position A to B' is performed. Starting point and target point of the product as well as rest at both points are identical for both processes. For other process types, this boundary condition may not be given.

The previously introduced motion processing approaches have different characteristics. In the case of discrete processing, it is assumed that there are areas where switching with non-optimal motion specifications has no negative influence on the process result. The process characteristics are not important in this case. The continuous switching principle can be used for processes where an interpolated value is close to the optimum value assigned to the given operating speed. Processes with basically different characteristics cannot be switched in this way. In this case, an excessive distortion of the interpolated motion compared to the optimal solution would induce instabilities in the process.

In the example processes, as described above, different physical principles are used to implement the technical task. As a result, the motion profiles at the working tool differ considerably. Depicted in Figure 8a are the X-components of the motions. Figure 8b shows the motion as a 2D plot in the workspace of the kinematics. Since the motion in Y is the same for both processes, it is not considered further here. When comparing the motions, the following is worth mentioning:

- Both motion profiles have an identical starting point in rest at $\tau = 0$ ($v = a = 0$). Discrete switching is allowed at this point.
- In the new process (light gray), the process stroke ($\tau = 0$ to 0.4) is in X direction significantly shorter than in the conventional process (dark gray). As a result, switching the process solutions within the process stroke would lead to a distortion of the resulting motion set-points due to the strongly differing motion profiles. This, in turn, could induce process instabilities, which, however, must be avoided. Switching within the process stroke is therefore not allowed for the example.
- The return stroke ($\tau = 0.4$ to 1) is also different for both process solutions. It can be seen that with the conventional solution (dark gray), the comb moves in a negative direction directly after the end point of the process stroke, while the new solution

(light gray) first moves in a positive direction. This is related to the positioning of the comb and the product. Whereas in the conventional solution the product is in contact with the comb in the positive X direction, in the new solution, it is positioned in the negative direction. To avoid a collision, the comb must first be moved in the other direction. Since the return stroke is not relevant to the process in the example, there is the possibility of switching between the motion profiles here. It is evident that the end points of the process stroke are far away from each other. Even here, there is no risk of collision when switching. The product is at the same position for both processes, and the working tool will not touch the product even in an interpolated motion. Switching with a weighting function is therefore permissible in the entire return stroke.

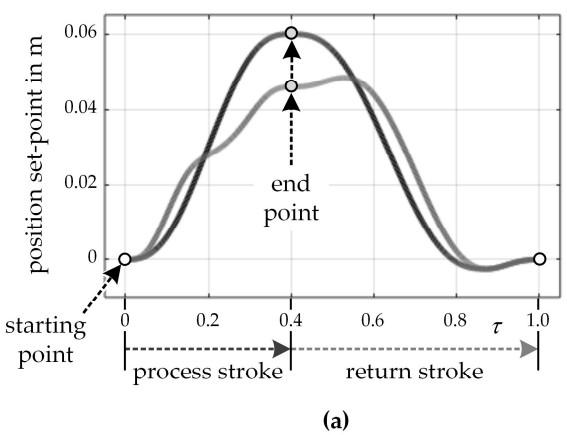

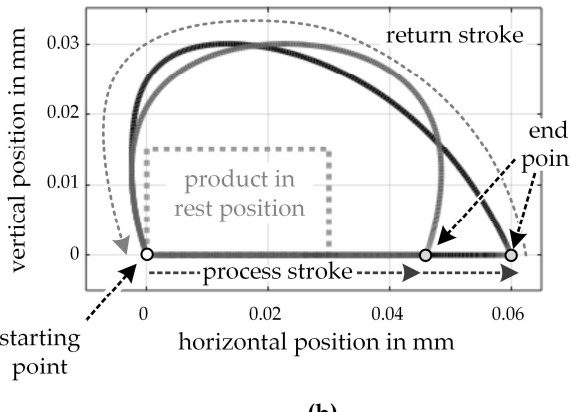

(a)    (b)

**Figure 8.** Comparison of the motion principles of the conventional approach (dark gray) and the new approach (light gray). (**a**) Motion set-point in horizontal direction and (**b**) resulting 2D motion path.

It should be noted at this point that the decision about which regions of the process are relevant or not must be made individually for each process. It cannot be generalized.

Discrete motion processing is successful in the given scenario when switching within the return stroke or at the common starting point. This has already been verified in various previous works [7,16,18,19]. It can thus be applied directly to the execution of the hybrid process discussed here.

### 5.3. Further Applications for Increasing the Operating Speed

Above, it is shown that when processing operation-speed-dependent motion profiles, different process solutions can be executed together. There is another aspect that should be addressed at this point. It was described and validated in the authors' preliminary works [7,16,18]. The idea is illustrated in Figure 9. By connecting more than two motion profiles, the process window can be further increased. In the lower speed range, the motion profile for the conventional process is used up to $f_{s1}$. There is only one motion profile in this range, which is used unchanged for all operating speeds. When the operating speed exceeds $f_{s1}$, the motion processing approach switches discretely to the new process solution, which is more suitable for higher operating speeds [7]. For our example, two motion profiles for the new process solution are shown. They are optimized for $f_2$ and $f_3$ (Figure 9, blue and green). If the operating speed is increased further, the processing approach again switches discretely to the third motion profile $f_3$ at operating speed $f_{s2}$. The operating speed is limited upwards at $f_{3.ucr}$ where the positioning error of $f_3$ exceeds the tolerable error. Combining three motion profiles extends the process window significantly compared to using individual motion profiles. The black dashed line in Figure 9 shows the expected motion error as measured in previous work [7,16,18].

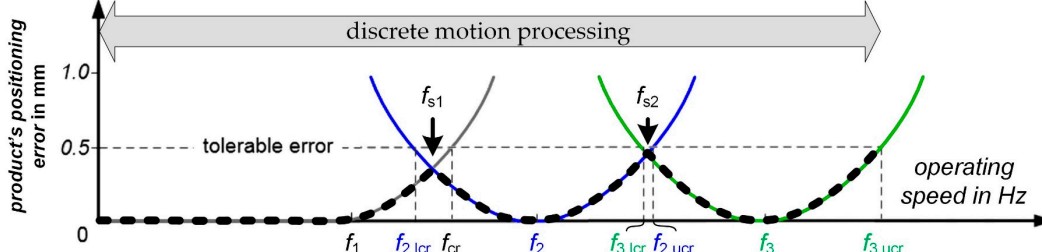

**Figure 9.** Principle of combining the two process solutions and three optimized motion profiles with the help of discrete processing at $f_{s1}$ and $f_{s2}$. The maximum operating speed can be increased compared to only using two motion profiles.

Figure 10 shows another method for increasing the operating speed even further. The idea is still based on discretely combining the different process solutions at $f_s$. In contrast to the approach discussed above (Figure 9), continuous processing and its advantages are used in the speed range above $f_s$.

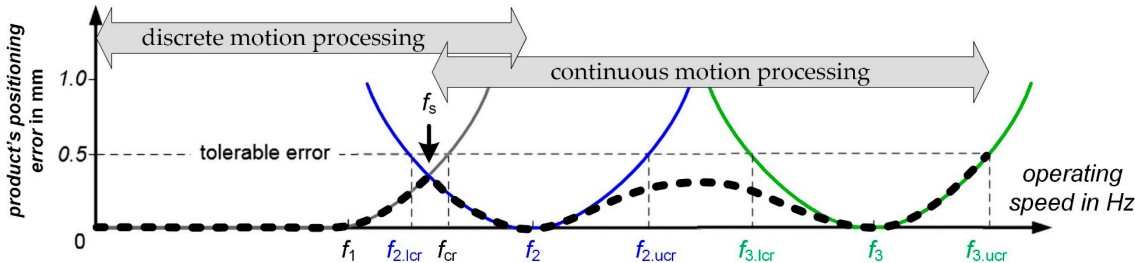

**Figure 10.** Principle of combining the two process solutions and three optimized motion profiles. Between the process solutions, discrete processing is used at $f_s$. Operating speeds above $f_s$ are processed continuously. The maximum operating speed can be further increased.

Below $f_s$, discrete processing of the conventional process is done. In the range between $f_s$ and $f_2$, processing can be either discrete or continuous, since both methods implement identical behavior. Between $f_2$ and $f_3$, continuous processing using 1D linear interpolation is performed. With continuous processing, the optimized motion profiles $f_1$ and $f_2$ can be further spaced than is possible with discrete processing [7,20]. This is because the motion error is reduced during the continuous processing compared to the discrete processing [16]. With enlarging the distance between $f_2$ and $f_3$, the error in between also increases. However, if the error remains below the tolerable error, the combination of those motion profiles for controlling the process is allowed. An algorithm that enables the identification of reasonable operating speeds $f_2$ and $f_3$ was presented in [20]. Although a different process example was used here, the algorithm can also be applied to the process considered here.

The black dashed line again shows the product's positioning error. The validity of the procedure and the appearance of the resulting error was evaluated in [7,16]. The operating speed can thus be increased up to the speed $f_{3.ucr}$.

By combining different process solutions with overlapping process windows, it is possible to enlarge the process window of the entire process. This means that the central requirement of being able to run a technological task from standstill to the highest possible operating speeds can be better met than with conventional control approaches. In the following chapter, guidance for deciding whether the approach is transferable to other processes is given.

*5.4. Applicability of the Approach on Other Processes*

The usage of the hybrid process solution was discussed using the example of an intermittent product transport. In this chapter, some further thoughts on using this new

approach with other processes will complete the discussion. This is intended to simplify decisions as to whether the approach is applicable with a process under consideration.

Technically, more than two different process solutions can be used together in this way. This could be done, for example, to change the process solution only in a small operation speed range (for example, to overrun a resonance). Within this range and on both sides beyond the range, different process solutions could be executed.

Expanding the process window upwards or downwards by combining different switching methods (discrete–continuous) is also possible in other combinations. For example, two continuously switchable process solutions could also be combined with discrete switching in between. The process windows only have to overlap favorably.

The following requirements should help to decide if the hybrid approach can be applied to other processes and process solutions.

- The hybrid approach can be used if both process solutions fulfill one and the same technological task. Combining different technological tasks is possible in principle but does not appear to be reasonable in the context of the discussion. Whether such a combination makes sense or not must be examined individually.
- The normalized cycle time $\tau$ of the motion profiles to be used in combination has to be synchronized; i.e., the process stroke has to be in the same $\tau$ range for all process solutions that have to be combined. If this requirement is not met, a synchronization solution for the relevant motion profiles must be considered, e.g., using an offset for different profiles.
- If the process solutions to be combined have common points at which the motion values ($s$, $v$, and $a$) are equal and located outside the process stroke, the hybrid approach can be used with the discrete switching principal. If the common points are within the process stroke, it must be carefully checked if switching is acceptable. Special attention has to be paid to the fulfillment of all given requirements. In the example of the transportation process, this would not be acceptable. If switching points exist at which the deviations are only small, they must be checked individually for the processes under consideration to determine how significantly the deviations influence the resulting process. Here, too, all given requirements have to be fulfilled, e.g., process stability.
- If there is a part of the motion profile that is not relevant for the process and its stability (e.g., a return stroke), it can be used for switching. The decision about which regions of the motion are not relevant must be made individually for each process under consideration. Furthermore, it is important to check carefully that no disturbances such as collisions between tool and product occur as a result of switching.

## 6. Summary

In this work, an approach was presented that enables combining process solutions with fundamentally different characteristics. Conventional motion processing approaches only support the execution of a single solution, and the combination of multiple process solutions is not allowed. Our approach is discussed using the example of conveying small-sized pieced goods in processing machines.

Two fundamentally different solutions for this process are introduced. The first solution is a conventional approach commonly used in industry. In this solution, a working tool shoves the product along the desired motion path. The advantage here is that this process solution can be executed even with very low operating speeds. Disadvantageously, the process is limited in its operating speed upwards by the physical principle used. In the second process solution, a fundamentally different physical principle is applied. Here, the product is transferred into a free-sliding phase in order to be decelerated later by the working tool. The advantage of this approach is that it is not limited upwards by the physical principle. The available mechanical components are the only limitation. Unfortunately, this process is limited in its operating speed downwards.

In this work, these two process solutions are combined in the hybrid motion processing approach. Using this approach, the advantages of the two processes are combined and the disadvantages are compensated. It is discussed that the process window of the hybrid approach can be much larger than the process window of a single solution. Subsequently, it is discussed which criteria other processes have to fulfill in order to be combinable with the hybrid approach. The presented approach is based on the previously developed and repeatedly validated knowledge of the processes and their realization. The validation of the approach on a real machine is the subject of the current research and will be presented in a future contribution.

**Author Contributions:** Conceptualization, O.H.; methodology, O.H.; validation, O.H. and C.T.; writing—original draft preparation, O.H.; writing—review and editing, O.H., C.T., S.I. and J.-P.M.; visualization, O.H. and C.T.; supervision, S.I. and J.-P.M. All authors have read and agreed to the published version of the manuscript.

**Funding:** This research was funded by the German Research Foundation under grant number 182157057.

**Institutional Review Board Statement:** Not applicable.

**Informed Consent Statement:** Not applicable.

**Acknowledgments:** The authors would like to thank the German Research Foundation for supporting this work under grant number 182157057.

**Conflicts of Interest:** The authors declare no conflict of interest.

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
