# Peer review of "Combining the Advantages of Different Processing Solutions Using a Novel Motion Processing Approach"

_applsci, doi:10.3390/app112110238_

Round 1
Reviewer 1 Report
The work is interesting and potentially valid for the current industrial scenario. However, some points need to be addressed carefully before it can be published. I suggest to spend time to fix these important issues before submitting again.
Comments:
1) The abstract should reflect the paper content. The current version also contains grammatical oddities. Please revise.
2) Section 1 must be re-worked. Some suggestions:
- Name it “Introduction” rather than “Motivation”.
- Although the use of >1 subsections improves the readability and clarity of the section, I think the actual layout needs to be improved. I would follow a typical schematic and try to point out clearly the motivations and the paper contributions. Bulleted lists would facilitate this task and guide the readers.
- The English writing must be improved in many parts. I suggest to proofread again (if possible, with the help of a native speaker).
3) A dense literature concerning the motion profile optimization of servo-drives employed in automated machines is available. Many concepts were already discussed in previous works and therefore need to be recalled appropriately.
4) The discussed approach is reported only qualitatively. A (real) case study must be included for validation. Quantitative discussions are needed.
5) Please increase the figures’ graphical quality when exporting the file.
6) The bibliography section is very poor. I suggest to include recent journal papers and to expand the discussion.
Author Response
Dear Reviewer,
thank you for your valuable feedback. Attached you can find a point-by-point response to your recommendations.
Best regards,
Olaf Holowenko

Reviewer 2 Report
This paper presents a novel motion control approach to combine incompatible process solutions in a hybrid process solution for increasing the machine’s achievable process window. Some simulated results are reported to verify the effect of the proposed method. To careful review the paper, a novel method for processing machines were proposed, however, a theoretical analysis should be detail for the proposed method in section 3. In particular, completed mathematical equations should be included for the proposed method. It is required that the illustrated examples should be clearly verified to understand the work of the authors. More information to show the effectiveness by using the new algorithm has also to be added. For example, the authors should describe the practical conditions of the experimental set-up, such that the readers can easily follow the proposed rule. Moreover, explanations and discussions on the results acquired between theoretical model and experimental readings should be given in the manuscript.
Author Response

(The authors gave the same response as above.)

Reviewer 3 Report
The presented article describes the application of a new and quite interesting approach to the analysis of the kinematics of motion of a one and strictly defined device used to move objects on the production line. However, in the course of analyzing the content of the article, the following comments can be made:
- There is a noticeable lack of description of the relationship between the rotational speeds of individual servo motors with each other and the time of one device operation cycle.
- The derivation and description of the dependence of the influence of geometrical values ​​of the kinematic system on the obtained speeds of the mechanism's actuator were omitted.
- The point on lines 287 - 292 indicates that the new advancement process is in the range τ = 0 to 0.4, although figure 8a shows that the proces stroke ends at approximately τ = 0.55.
- The new approach proposed in chapter 3.3 is described only on the theoretical level, and its legitimacy should be underpinned by a mathematical, physical model or the results of real tests.
Moreover, it should be pointed out that carrying out the analysis of the movement of the sliding object limited to only one case of the kinematic system narrows down the analysis of the presented problem. The described process can be analyzed, which the authors partially do, as two separate processes, accelerating the sliding object and its subsequent deceleration. In the analyzed, quite special case, the emphasis was placed on the optimization of operating-speed-dependent motion based only on the process speed expressed in Hz. However, it is possible, based on kinematic relationships or design changes, to optimize the process based on an independent selection of acceleration and deceleration parameters of the displaced object. Moreover, the article is a compilation of the knowledge already presented in previous publications.
Author Response

(The authors gave the same response as above.)

Reviewer 4 Report
Review Report of applsic-1408649
In this paper, a novel motion processing approach with the combination of different process solutions is proposed for increasing the machines’ achievable process window. The proposed approach can fully merge the advantages and compensate for the disadvantages of different process solutions, and the work has practical implications. However, the authors must carefully address the following suggestions in an improved manuscript.
General comments
#1: In terms of novel contributions. The contributions of this paper have not been reflected clearly. The paper should add more introductions to the approach proposed, and the highlights should be obviously presented.
#2: In terms of method validation. The proposed approach is explained to be feasible and can increase the operating speed in Section 3.2 and Section 3.3. However, a more comprehensive comparison with other methods is recommended (if any) to better explain the superiority of the proposed method.
Specific comments:
#1: Has line 313 “The idea is illustrated in Figure 9” been fully stated?
#2: The meaning of curves with different colors should be explained clearly in all figures, especially in Figures 7-10.
#3: In Section 3.2 and Section 3.3, the validation and application are mainly presented by statements. A supplement of useful calculated data can enhance the persuasiveness of the proposed approach.
#4: In Section 4, the highlights are also not emphasized. The highlight and major contributions of this manuscript should be clearly presented instead of lengthy statements, and valuable data, which can show the superiority of the novel approach should be presented.
Author Response

(The authors gave the same response as above.)

Round 2
Reviewer 1 Report
The authors have addressed my comments. I only suggest to change the title of Sec. 2.
Author Response
Dear reviewer,
the title of Section 2 has been changed.
Thank you for reviewing the article!
Olaf Holowenko
Reviewer 3 Report
Thank You for Your response. The decision rests with the editorial office now.
Author Response
Dear reviewer,
Thank you for reviewing the article!
Olaf Holowenko
Reviewer 4 Report
Overall, the paper has been revised well in this round, however, after reading the revised version, a slight comment should be considered: The abstract of this paper is not sufficient. Overall, the Abstract is too long and the background is not mandatory, it is even suggested to start the abstract from “In this article…”. In the abstract, the authors can only provide the method they have proposed, the primary results, and the significance of the methodology.
Author Response
Dear reviewer,
as suggested, the abstract has been shortened and clarified.
Thank you for reviewing the article!
Olaf Holowenko